# Ancient DNA of the Don-Hares Assumes the Existence of Two Distinct Mitochondrial Clades in Northeast Asia

**DOI:** 10.3390/genes14030700

**Published:** 2023-03-12

**Authors:** Fedor Sharko, Natalia Slobodova, Eugenia Boulygina, Maksim Cheprasov, Maria Gladysheva-Azgari, Svetlana Tsygankova, Sergey Rastorguev, Gavril Novgorodov, Gennady Boeskorov, Lena Grigorieva, Woo Suk Hwang, Alexei Tikhonov, Artem Nedoluzhko

**Affiliations:** 1Research Center of Biotechnology of the Russian Academy of Sciences, 119071 Moscow, Russia; 2Kurchatov Center for Genomic Research, National Research Centre “Kurchatov Institute”, 123182 Moscow, Russia; 3Faculty of Biology and Biotechnology, HSE University, 101000 Moscow, Russia; 4Lazarev Mammoth Museum, M.K. Ammosov North-Eastern Federal University, 677000 Yakutsk, Russia; 5Federal Research Centre “The Yakut Scientific Centre of the Siberian Branch of the Russian Academy of Sciences”, 677980 Yakutsk, Russia; 6Laboratory of Experimental Embryology, Institute of Translational Medicine, Pirogov Russian National Research Medical University, 117997 Moscow, Russia; 7Institute of Diamond and Precious Metals Geology, Siberian Branch of the Russian Academy of Sciences, 677007 Yakutsk, Russia; 8Center of Molecular Paleontology, M.K. Ammosov North-Eastern Federal University, 677000 Yakutsk, Russia; 9UAE Biotech Research Center, Abu Dhabi 30310, United Arab Emirates; 10Department of Biology, North-Eastern Federal University, 677000 Yakutsk, Russia; 11Zoological Institute of the Russian Academy of Sciences, 190121 Saint Petersburg, Russia; 12Paleogenomics Laboratory, European University at Saint Petersburg, 191187 Saint Petersburg, Russia

**Keywords:** Don-hare, *L. tanaiticus*, mountain hare, *L. timidus*, Holocene, ancient DNA, mitogenome, sequencing, extinction, Pleistocene-Holocene transition

## Abstract

Paleoclimatic changes during the Pleistocene–Holocene transition is suggested as a main factor that led to species extinction, including the woolly mammoth (*Mammuthus primigenius*), Steller’s sea cow (*Hydrodamalis gigas*) and the Don-hare (*Lepus tanaiticus*). These species inhabited the territory of Eurasia during the Holocene, but eventually went extinct. The Don-hare is an extinct species of the genus *Lepus* (Leporidae, Lagomorpha), which lived in the Late Pleistocene–Early Holocene in Eastern Europe and Northern Asia. For a long time, the Don-hare was considered a separate species, but at the same time, its species status was disputed, taking into account both morphological data and mitochondrial DNA. In this study, mitochondrial genomes of five Don-hares, whose remains were found on the territory of Northeastern Eurasia were reconstructed. Firstly, we confirm the phylogenetic proximity of the “young” specimens of Don-hare and mountain or white hare, and secondly, that samples older than 39 Kya form a completely distinct mitochondrial clade.

## 1. Introduction

The reconstruction of ancient ecosystems plays an important role in assessing many characteristics of the “mammoth steppe” (Arctic steppe), which has no analogues in modern times. To date, the evolution of the late Pleistocene megafauna species has been described in depth, while the population structure and demographic history of lagomorphs are poorly understood. Moreover, the systematic status of almost all small mammals of the Late Pleistocene on the territory of Russia is almost unexplored [1,2,3].

The small mammals from the Rodentia and Lagomorpha orders are widely used as a key indicator of ecosystem dynamics and environmental change. Moreover, historically, rodents have been actively implicated in evolutionary and ecological studies [4]. At the same time, despite the recent success, our knowledge of the origin, biogeography, and evolution of Lagomorpha is still limited [5].

The modern Lagomorpha order includes the Ochotonidae (pikas) and Leporidae (rabbits and hares) families [6]. The former are stenobionts (i.e., inhabit specific ecological niches), while the latter occupy a broad range of landscapes across all continents (except Antarctica) and adapt to different environmental conditions [6,7].

The Lepus genus consists of more than 30 species, whose common ancestor most likely originated in North America [8] approximately five million years ago [9].

Nowadays, only four hare species inhabit northern Eurasia: brown hare (*L. europaeus*), mountain hare (*L. timidus*), Manchurian hare (*L. mandshuricus*), and Tolai hare (*L. tolai*). Previous studies demonstrated the existence of ancient genomic introgression between *Lepus* species, particularly between *L. timidus* and *L. europaeus* [9,10,11,12]. Moreover, the hybridization between those species might have an adaptive effect [9,11].

The Don-hare is an extinct species of the genus *Lepus* (Leporidae, Lagomorpha) whose remains were first described at the Kostenki Upper Palaeolithic sites located on the banks of the Don River [13]. *L. tanaiticus* body size exceeds that of most of the modern mountain hare populations except the largest Arctic subspecies of *L. timidus tschuktschorum*, which approximates the smallest samples of *L. tanaiticus* [14]. This species has a lower jaw with a high dental part; in the third premolar (P3), the height of the lower jaw is much higher than that of *L. timidus*. In general, the structure of the molars of Don hares is similar to those of modern mountain hares, but they are very massive, like the other parts of the skeleton [15]. The Don-hare had large and more complex teeth and powerful jaws than the modern mountain hare, obviously feeding on coarse feeds, whose composition has not yet been determined to date [13,16].

The taxonomic status of the Don-hare is still debatable. Some authors presume that *L. tanaiticus* cannot be an ancestor of the modern Arctic hare (*L. arcticus*) [16], while others dispute even the independent species status of the Don-hare [17]. To solve this problem, it is desirable and necessary to have sufficient chronologically constructed material from the Late Pleistocene together with the use of modern ancient DNA methods.

Museum specimens are being used more and more often in evolutionary and conservation biology for solving fundamental and applied problems using a novel arsenal of molecular genetic techniques. DNA isolation from museum specimens and high-capacity DNA sequencing opens up new opportunities, associated with the investigation of the extinct or endangered species’ evolution and phylogeny [18,19,20,21,22]. In the previous study, comparative phylogenetic analysis of *L. tanaiticus* and *L. timidus* specimens, based on a 338-base pair (bp) fragment of the D-Loop of the mitochondrial genome, has not supported the assumption that *L. tanaiticus* and *L. timidus* belong to different species [23].

In the present study, we carry out a comparative mitogenomic analysis of the partial (*D-Loop*) and whole mitochondrial genome of Pleistocene Don-hare fossils (28,360 ± 170–50,120 ± 1210 years BP) from northeastern Eurasia (Yakutia, Russia) and other ancient and modern hare specimens.

We demonstrate that there were at least two mitochondrial clades of hares in Northeast Asia. Our data again calls into question the taxonomic status of the Don-hare. We assume that the ancient mitochondrial clade (which was older than 39 Kya) was replaced by a more modern mitochondrial clade, apparently obtained by the Don-hare from the mountain hare during possible introgressive hybridization. Nevertheless, subsequent studies on the nuclear genomes, as well as the involvement of *L. timidus tschuktschorum* specimens, are necessary to clarify the taxonomic status of the Don-hare.

## 2. Materials and Methods

### 2.1. Don-Hare Specimen Description and ^14^C Dating

Four ancient arctic Don-hare specimens from three permafrost sites in northeastern Siberia (Yakutia, Russia) are stored at the Laboratory of P. A. Lazarev Mammoth Museum of the M.K. Ammosov North-Eastern Federal University. A sample with museum number GM-7133 is stored at the Institute of Diamond and Precious Metals Geology (Figure 1). All these specimens were examined in this study (Figure 2; Table 1).

The body mass of the Don-hares considered in the study (Figure 2A–E) could not be measured with high accuracy because all the specimens have undergone mummification and cold desiccation. Their initial mass has significantly decreased over thousands of years in the Yakutian permafrost. For example, the mummy of the Don-hare from the Ogorokha River (Z43 specimen) weighs 1.2 kg, and based on its body size (by analogy with modern mountain hares), during his lifetime it could weigh 3.5–4 kg.

A young Don-hare specimen from Yana River (Z1 specimen; apparently *subadultus*) has a body length of 480 mm (Figure 2A). This Z1 specimen’s body length is comparable to the minimal length for modern adults of mountain hares from central Yakutia [24]. The adult Don-hare specimen from Ogorokha River (Z43 specimen) has a body length of 642 mm (Figure 2E), which is close to the average body length of the Chukchi white hare (*L. t. tschuktschorum*), which is known as one of the largest subspecies of the modern mountain hare [25].

According to preliminary morphological analysis, the Don-hare specimens used in this study were large individuals. The length of the tibia measured without the destruction of the mummies is about 150 mm in the Z1 specimen (Figure 2A) and about 145 mm in the Z3 specimen (Figure 2B). In the Z43 specimen (Figure 2E), the tibia has a length of 145 mm. This length of the tibia exceeds the maximum length of the tibia in modern mountain hares, 144 mm (Boeskorov, unpublished data). The hare specimens analyzed correspond to the Don-hare morphology based on the main diagnostic traits of this species. Don-hare specimens were distinguished by a large height of the lower jaw in front of P3: Z1—16.6 mm, Z4—17 mm, and Z43—16.6 mm, which exceeds the height of the lower jaw in front of P3 in *L. timidus* (including *L. t. tschuktschorum* subspecies). The studied Don-hare specimens are also identified by a large diastema length: Z1—24.6 mm, Z4—22.1 mm, Z43—20.7 mm as well as the remarkable length of the third premolar (P3): Z1—4.6 mm, Z4—3.9 mm, and Z43—4.1 mm [13,16].

Three specimens (Yn-3/16, MM-F29, and MM-F51) were radiocarbon dated at the Carbon Analysis Lab Co., Ltd. (Gyeryong, Republic of Korea), and one specimen (MM-F53) at the Unique Scientific Facility «AMS BINP SB RAS» (Novosibirsk, Russia) was dated based on acid-based collagen extracted fraction. The acid–alkali–acid method with HCl and NaOH was used to remove carbonate and any excess contamination from the specimens [26]. The collagen was ^14^C dated by Accelerator Mass Spectrometry (AMS) (Table 1; Appendix A).

**Table 1 genes-14-00700-t001:** Nomenclature and sampling location of the Don-hare specimens used in this study.

Museum ID	Laboratory ID	Tissue Type	Year of Sampling	Specimen Origin	Specimen	AMS ^14^C Date, BP
Yn-3/16	Z1	skeletal muscle	2016	Yakutia, Verkhoyansk District, Yunyugen, middle section of the Yana River	Animal carcass	28,360 ± 170 ^1^
MM-F29	Z3	cartilage	2016	Yakutia, Verkhoyansk District, Yunyugen, middle section of the Yana River	Back limb of the animal	28,360 ± 170 ^1^
MM-F51	Z4	skin with hairs	2019	Yakutia, Abyysky District, Ayan, Badyarikha River	Mummified head of the animal	50,120 ± 1210
GM-7133	Z43	skeletal muscle	2021	Yakutia, Abyysky District, Ogorokha river	Partial animal carcass	>45,000 ^2^
MM-F53	Z5	skin with hairs	2020	Yakutia, Nizhnekolymsky District, Kolyma River downstream	Partial animal carcass	39,300 ± 529

^1^ Specimens Yn-3/16 and MM-F29 were found in the same Yunyugen paleontological site. ^2^ This carcass was found at the same location where the skull of a sable (*Martes zibellina*) was previously found and dated >45,000 BP (Gra-62462) [27].

The Don-hare carcass (museum number: GM-7133) was found on the Ogorokha River at the same location where the skull of a sable was previously recovered and dated >45,000 BP (Gra-62462) [27]. Nevertheless, the preservation of its skull and its weak mineralization indicates that it belongs to the Karginian interstadial of the second half of the late Pleistocene, which began 60–65 Kya years ago. Several radiocarbon dates of Eurasian cave lion (*Panthera spelaea*) of Karginian (30,900 ± 390, 30,970 ± 380 years ago) and presumably the Karginian interstadial (>47,600, >49,800 years ago) were also obtained from the location of the Ogorokha River paleontological site [28].

### 2.2. Ancient DNA Isolation, DNA Library Preparation and Sequencing

Each specimen was mechanically cleaned of soil residues and treated with ultraviolet (UV) light for 15 min before DNA extraction. Genetic material was extracted from the Don-hare specimens in the special clean room of the ancient DNA complex in the National Research Center “Kurchatov Institute” (Moscow, Russia) using a silica beads-based DNA isolation protocol [29,30] which contains five main steps:(1)Teeth drilling for powder.(2)Teeth powder digestion in buffer containing EDTA and proteinase K.(3)DNA enrichment using silica beads in the binding buffer which contains tris(hydroxymethyl)aminomethane (TRIS), sodium acetate and sodium chloride, and guanidine thiocyanate.(4)DNA extracts washing in ethanol.(5)DNA elution from the beads.

Five independent ancient DNA extractions (repeated 3 times per specimen) were carried out. The DNA quantity and quality were determined with Qubit 3.0 (Thermofisher Scientific, Waltham, MA, USA). Five multiplexed DNA libraries were prepared from the extracts with the highest DNA concentration using an Ovation^®^ Ultralow Library System V2 kit (Tecan Group Ltd., Männedorf Switzerland). DNA libraries were constructed in ancient DNA facilities, final amplification of the DNA libraries was performed in the modern DNA facility of the National Research Center “Kurchatov institute” (Moscow, Russia). The amplified DNA libraries were quantified using a high-sensitivity chip on a 2100 Bioanalyzer instrument (Agilent Technologies, Santa Clara, CA, USA). Multiple negative controls were used during the ancient DNA extraction and DNA library amplification. The negative controls did not contain DNA after the DNA extraction, and the DNA libraries, which were prepared from the negative controls, were not amplified. Controls were not used for subsequent DNA sequencing.

Five DNA libraries were sequenced on the S2 flowcell of the Illumina Novaseq6000 genome analyzer (Illumina, San Diego, CA, USA) at the sequencing facility of Kurchatov Center for Genome Research (Moscow, Russia) with paired-end reads of 150 bp in length.

### 2.3. Bioinformatics Analysis

Raw DNA reads were converted to FASTQ format with bcl2fastq (v2.20): https://support.illumina.com/downloads/bcl2fastq-conversion-software-v2-20.html (accessed on 18 January 2022). Sequencing data trimming was carried out with default parameters using the AdapterRemoval2 tool (version 2.2.2) [31]. Sequencing data were processed through the PALEOMIX 1.2.14 pipeline [32], including the merging of paired reads and mapping using BWA v0.7.17 with “rescale” option steps [33]. Mapping was done against the mountain hare reference genome sequence (GCA_000003625.1, CIBIO-ISEM_LeTim_1.1, Mountain hare assembly). Postmortem DNA damage patterns were analyzed using the MapDamage v2.0 tool [34]. We used the MapDamage models to downscale the base quality scores according to the probability of DNA damage from by-products to reduce the impact of nucleotide misincorporations in the downstream analyses. Three nucleotides from each end of each DNA read were trimmed using trimBam from bamUtil repository [35].

Filtered genomic data were used for de novo assembly of the mountain hare mitochondrial genomes using SPAdes v3.15.3 [36]. The mitochondrial DNA sequences of *L. tanaiticus* were extracted from the genome assemblies using blastn 2.7.1+ and an NCBI database. The resulting mitogenome sequences were annotated using MITOS [37]. The obtained annotations were then used to define partitions in the subsequent phylogenetic analysis.

The 338-base pair (bp) fragments of the D-Loop of the Don-hare mitochondrial genome, which was previously used for genetic analysis of specimens from the Pymva Shor I (Polar Urals, Russia) [23], were used for the comparative mitogenome analyses of specimens from Yakutia (this study) together with modern specimens of mountain hare—*L. timidus*, Arctic hare—*L. arcticus*, tolai hare *L. tolai*, Hainan hare—*L. hainanus*, Chinese hare—*L. sinensis*, woolly hare—*L. oiostolus*, Yarkand hare—*L. yarkandensis* and brown hare—*L. europaeus*. The European rabbit (*Oryctolagus cuniculus*) and the Northern pika (*Ochotona hyperborea*) were used as an outgroup (Appendix A).

Phylogenetic analysis of the coding sequences (CDS) of mitochondrial DNA was performed for the Eurasian specimens of the *Lepus* genus (including Don-hare). The European rabbit and Northern pika mitochondrial genomes were also used as outgroups (Appendix A).

The multiple sequence alignment was obtained with a ClustalW using gap open penalty (1500) and gap extension penalty (6) parameters [38]. The maximum likelihood (ML) analysis was conducted using RAxML v8.2.12. The maximum parsimony analysis was performed using the Subtree-Pruning-Regrafting model with 100 iterations of bootstrap testing in RAxML [39]. Phylogenetic tree reconstructions for D-Loop sequences and CDS were drawn in iTOL v4 [40].

## 3. Results

### 3.1. AMS Radiocarbon Dating

The Pleistocene Don-hare specimens, which were collected in Northern East Russia (Figure 1; Table 1), are stored in P.A. Lazarev Mammoth Museum (Yakutsk, Russia). The tissues revealed a finite age from 28,360 ± 170 up to 50,120 ± 1210 BP (Table 1; Appendix A).

### 3.2. Ancient DNA Sequencing and Don-Hare Mitogenomes Assembly

The total number of DNA reads generated for five Don-hare DNA libraries varied from 3,655,655 to 192,185,219 per library. The number of endogenous reads was measured for each DNA library with the PALEOMIX v1.2.14 pipeline [32]. The endogenous DNA content varied from 9.19 to 59.40 percent among the DNA libraries (Appendix A).

DNA reads obtained during the sequencing run were merged and then used for de novo assembly of the mitogenomes of the Don-hares. The mitogenome sequences of *L. tanaiticus* were extracted from the assembled contigs using blastn 2.7.1+. The mitogenomes of the Don-hares studied consist of a sequence between 16,343 to 17,147 bp (GenBank accession numbers: OQ270737–OQ270741) and includes 22 tRNA, 2 ribosomal RNA and 13 protein-coding genes (Figure 3A).

The mitochondrial genome assemblies of Don-hare contain a conserved number of mitochondrial genes and the typical gene order, codon usage and base composition, which are common among other vertebrates (Figure 3A). The base composition of the coding sequence (CDS) in the mtDNA descending order was 33%—T, 29%—A, 25%—C, and 13%—G with 38% of GC content. All five mitochondrial genomes of the Pleistocene Don-hares have the same start and stop codons. Ten (*ATP6*, *ATP8*, *COB*, *COX1*, *COX2*, *COX3*, *NAD1*, *NAD4*, *NAD4L*, *NAD6*) of the 13 protein-coding genes used ATG as a start codon; another two (*NAD2*, *NAD3*) used ATT; and *NAD5* has an ATA start codon. Nine genes (*ATP6*, *ATP8*, *COX1*, *COX3*, *NAD2*, *NAD4*, *NAD4L*, *NAD5*, *NAD6*) ended with a TAA stop codon, and another four (*COX2*, *NAD1*, *NAD3*, *NAD6*) use a TAG stop codon. The same start and stop codons are presented in the mitochondrial genome of the mountain hare [41].

### 3.3. Phylogenetic Analysis of Pleistocene Don-Hares from Northeast Asia Based on Their Mitochondrial Sequences

To describe the phylogenetic relationships of the Don-hare specimens among other *Lepus* species, we generated a phylogenetic hypothesis based on whole mitochondrial DNA sequences and maximum parsimony methods. Mitochondrial DNA sequences indicate *L. tanaiticus*, *L. timidus*, *L. tolai*, *L. tibetanus* and *L. arcticus* are clearly distinct from other *Lepus* species, such as *L. yarkandensis*, *L. sinensis* or *L. europaeus.* Interestingly, the mitogenomes of “old” *L. tanaiticus* specimens (dated >39,000 BP) cluster separately from the “young” specimens (dated <30,000 BP) (Figure 3B).

We also evaluate the genetic distances of Yakutian Pleistocene Don-hares with previously published data from the Polar Urals (dated by 3–6 Kya BP) [23] using a 338-bp D-loop region. The phylogenetic reconstruction shows that Don-hares are clustered in two subclusters. One of them consists of the “young” *L. tanaiticus* together with the Don-hares from Polar Urals and modern Arctic and mountain hares and the other consists of “old” *L. tanaiticus* and *L. yarkandensis* (Figure 4). However, the results of phylogenetic reconstruction based on the D-loop fragment are not confident due to low bootstrap values which indicate that there is a conflicting signal in the phylogenetic tree.

## 4. Discussion

The Don-hare (*L. tanaiticus*) is described as very large, exceeding (or corresponding to) the body size of the largest modern specimens of the mountain hare. The taxonomical status of this species is still debatable after more than 50 years of research [13]. Some authors distinguish it as a separate species [16], while others consider it a morphotype of modern mountain hares [17]. Recent studies based on a non-coding sequence of a D-Loop fragment of the mitochondrial genome have posed a question about the systematic status of *L. tanaiticus* [23].

Here, we tried to evaluate the phylogenetic relationship of the Don-hare using complete mitochondrial genomes of *L. tanaiticus*, as well as the mitogenomes of modern species of the *Lepus* genes. We have shown a phylogenetic relationship between the species *L. tanaiticus*, *L. timidus*, *L. tolai*, *L. tibetanus* and *L. arcticus*, which suggests their genetic kinship as well as the possibility of introgressive hybridization between them. Interestingly, *L. tanaiticus* specimens cluster separately according to their age, which suggests the existence of at least two mitochondrial clades of *L. tanaiticus*.

Over the last few decades, a large body of evidence has accumulated that directly demonstrates the importance of hybridization between closely related species and the exchange of alleles as one of the driving facts of evolution, allowing not only the exchange of adaptive traits [42,43,44] but also causing explosive speciation [18,45]. Recent studies have suggested that hares represent a widespread radiation of species characterized by striking ecological adaptations and recurrent admixture [9,46]. Based on our preliminary mitochondrial genome data, we speculate the existence of ancient hybridization between *L. tanaiticus* and *L. timidus* at the end of the Late Pleistocene; the same as described for *L. timidus* and *L. europaeus* [9,11,12].

On the other hand, these results represent the fact that Pleistocene faunal diversity has not been fully understood, and future morphological, isotope and genome-based studies of specimens from the melting Siberian permafrost will allow us to describe novel and previously unexplored Lagomorpha species.

Finally, we assume that more in-depth studies of the dated series of *L. tanaiticus* specimens and their nuclear genomes, as well as the nuclear genomes of modern and historical *L. timidus tschuktschorum* specimens, will shed light on the evolution of the *Lepus* genus in Northeast Eurasia during the Pleistocene–Holocene transition.

## Figures and Tables

**Figure 1 genes-14-00700-f001:**
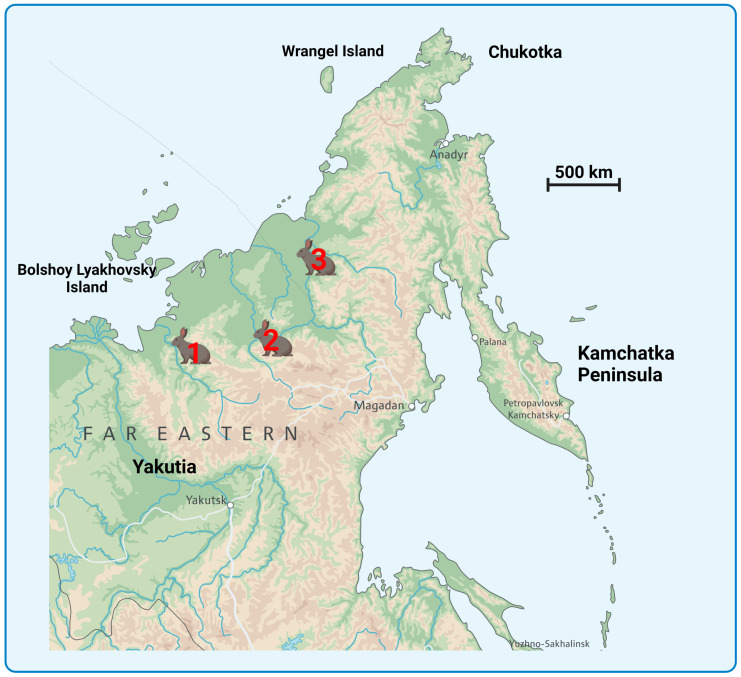
Map showing the sampling sites of Don-hare specimens used in this study. The Don-hare specimen description is presented in Table 1. (1) Specimens Z1 (Yn-3/16) and Z3 (MM-F29); (2) Specimens Z4 (MM-F51) and Z43 (GM-7133); (3) Specimen Z5 (MM-F53).

**Figure 2 genes-14-00700-f002:**
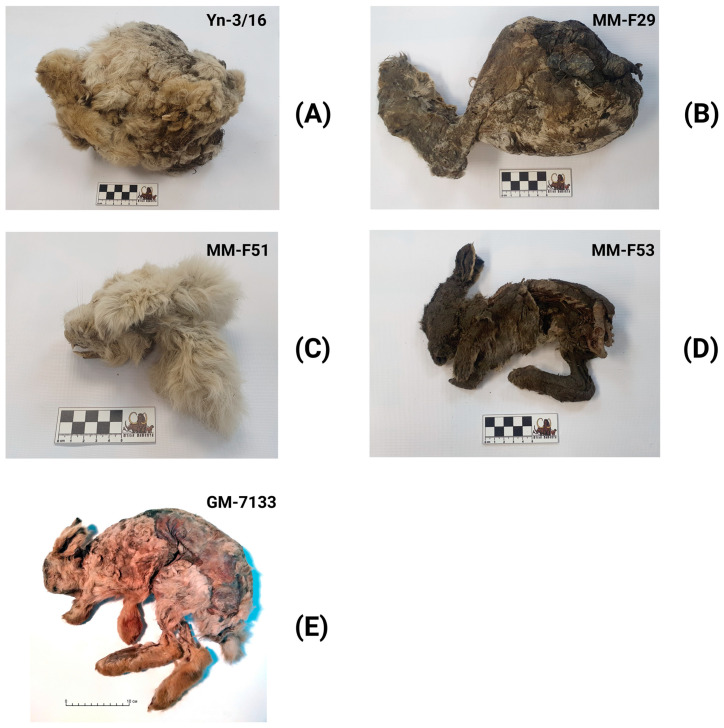
Frozen mummies of Don-hare used in this study. The Don-hare specimen description is presented in Table 1. (**A**) Specimen Z1 (Yn-3/16); (**B**) Specimen Z3 (MM-F29); (**C**) Specimen Z4 (MM-F51); (**D**) Specimen Z5 (MM-F53); (**E**) Z43 (GM-7133).

**Figure 3 genes-14-00700-f003:**
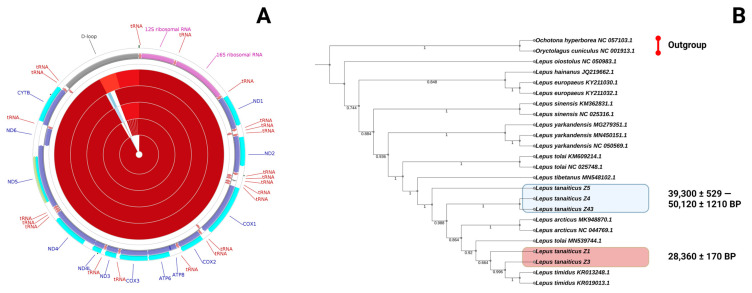
The mitochondrial genomes of Pleistocene Don-hare. (**A**) Circular representation of the *L. tanaiticus* mitochondrial genomes (specimens Z1, Z3, Z4, Z43 and Z5). Genes encoded by the heavy (H) strand are shown outside and those encoded by the light (L) strand are shown inside the mitogenome. Different genes are shown as filled boxes in different colours. (**B**) Parsimony phylogenetic tree of *Lepus* species, including the “young” (marked by the red box) and “old” (marked by the blue box) *L. tanaiticus* specimens, based on their mitochondrial DNA sequences. Bootstrap values are shown in the nodes of phylogenetic reconstruction.

**Figure 4 genes-14-00700-f004:**
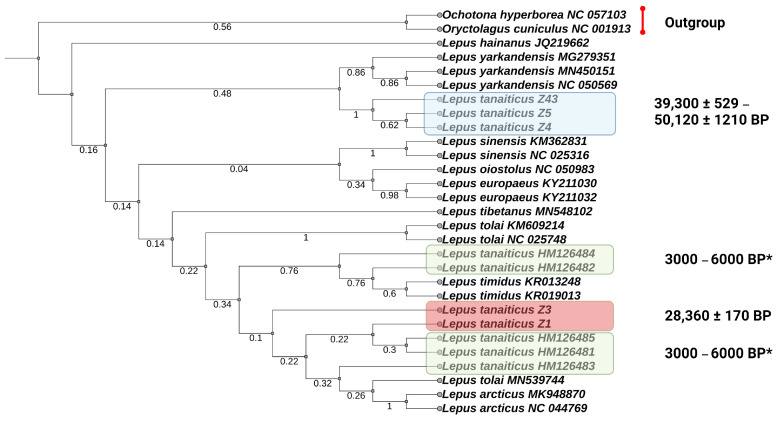
Parsimony phylogenetic tree of the *Lepus* species, including the extinct Don-hare based on nucleotide variability of a 338 bp D-Loop fragment. “Young” Yakutian specimens of *L. tanaiticus* from this study are marked by a red box and “old” Yakutian specimens of *L. tanaiticus* from this study are marked by a blue box. Green boxes represent previously published *L. tanaiticus* specimens from the valley stream Pymva Shor (Polar Urals, Russia) [23]. *—These individuals are dated as Holocene specimens according Prof. Stefan Prost.

## Data Availability

The mitochondrial genome assemblies of fossil hares published in this study are available for download through the National Center for Biotechnology Information (NCBI): OQ270737–OQ270741. Accession numbers of previously published *Lepus* specimens that were implemented in this study are available in Appendix A (mitochondrial D-Loop sequences) and Appendix A (mitogenome sequences).

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
