# Peer review of "Ancient DNA of the Don-Hares Assumes the Existence of Two Distinct Mitochondrial Clades in Northeast Asia"

_genes, 2023, doi:10.3390/genes14030700_

Round 1

Reviewer 1 Report

In the presented manuscript, Fedor Sharko and coworkers have studied the mtDNA lineages in extinct hare species using ancient DNA. The study is methodologically sound and well conducted. The paper is well written and structured, with only minor issues with language. I have, however, a few suggestions to improve the presentation and interpretation of the data:

Introduction

L75: Are there any estimates for the body mass of the Don hare (L. tanaiticus)? Because these species are quite unfamiliar to most readers, it would be useful to have a mass comparison with this species vs the mountain hare (L. timidus timidus) and Chukchi hare (L. t. tschuktschorum). Also, any other relative comparisons of body part sizes (tibia length etc) would be helpful. What I read from my Ognev (1940, p. 258) tschuktschorum has quite large skull, length up to 116 mm, width 57 mm, which seems comparable with the Don hare. He has also illustrated the skull, which seems quite different from the nominal species. However, as Gureev (1964) illustrated only fragments, it is difficult to get an impression of the morphological differences. Therefore, if the authors can provide more detail helping to differentiate Don hare from the Chukchi hare, this would be very valuable.

L77: “high dental part” needs to be more specific. Reading from Gureev (1964): […] jaw with high dentition, especially the P3 (16.0-19.8 mm) and M3 (18.5-20.2 mm) [vs 13.4-14.2 mm and 15.3-18.2 in timidus). However, the teeth structure is apparently very similar to timidus? 

Materials and methods: 

L107-125: very impressive finds. It would be great to have some images of these specimens to accompany the text. Morphological comparison of the dentition would be interesting for the specimens with skulls, at least for P3 and M3. These should be compared with contemporary specimens of the Chukchi hare.

L173-4: If the mapping of the sequencing data was done against the whole mountain hare genome, why was only the mtDNA used for the analyses?

Results:

L236: See above. How certain it is that the young specimens represent the Don hare and not the Chukchi hare? What is the identification of the specimens based on?

Discussion:

Very nice. However, the main issue with studies on extinct species is how to obtain certainty with the species identification. I believe this is evident from the fact that different “tanaiticus” specimens cluster all around the Lepus phylogenetic tree (Figure 3). The only way to solve this would be to sequence the original type specimen, which is probably not plausible and therefore I do not require it for this study. Unfortunately, Gureev (1964) does not give much information of their age or context but considering that the type locality is in Southern Urals (Novgorod-Seversky, Voronezh region), I would not be surprised if these turn out to be something different. I think that it would be worthwhile to discuss this uncertainty in broader detail. While hybridization (L275) is plausible and known to occur, I believe that the genetic evidence shows ancient species diversity, which has not been fully understood in morphological studies and probably contains unknown species, which hopefully the nuclear genome data will someday elucidate. Unless the authors have other evidence, I would interpret that the robust animals, which cluster in the same clade with timidus (HM12682, 84) can well represent the the Chukchi hare, specimens Z4, Z5, Z43 (I hope that the letter Z has no political implications here) a species related to yarkandensis and Z1, Z3, HM126481, 83 and 85 a third species. It would be very informative to include the ages of the other “tanaicus” specimens as well as their geographic locations into the figure 3. Regardless if the authors agree with my interpretation or not, the implications of the phylogenetic tree needs to be discussed in more detail.

Minor issues:

Abstract: remove "kilo years ago".

Reference:

Ognev SI (1940) Mammals of the USSR and adjacent countries. Vol 4. Akademiya Nauk SSSR, Moscow. pp. 615.

Author Response

We would like to thank you for your consideration, valuable comments, and suggestions to improve this manuscript. Corrections in manuscript are marked by yellow. Answers are in blue.

In the presented manuscript, Fedor Sharko and coworkers have studied the mtDNA lineages in extinct hare species using ancient DNA. The study is methodologically sound and well conducted. The paper is well written and structured, with only minor issues with language. I have, however, a few suggestions to improve the presentation and interpretation of the data:

Introduction

L75: Are there any estimates for the body mass of the Don hare (L. tanaiticus)? Because these species are quite unfamiliar to most readers, it would be useful to have a mass comparison with this species vs the mountain hare (L. timidus timidus) and Chukchi hare (L. t. tschuktschorum). Also, any other relative comparisons of body part sizes (tibia length etc) would be helpful. What I read from my Ognev (1940, p. 258) tschuktschorum has quite large skull, length up to 116 mm, width 57 mm, which seems comparable with the Don hare. He has also illustrated the skull, which seems quite different from the nominal species. However, as Gureev (1964) illustrated only fragments, it is difficult to get an impression of the morphological differences. Therefore, if the authors can provide more detail helping to differentiate Don hare from the Chukchi hare, this would be very valuable.

Thank you for the comment. This text was added to the Material section (L139-L146): “The hare specimens analyzed correspond to the Don-hare morphology based on the main diagnostic traits of this species. Don-hare specimens were distinguished by a large height of the lower jaw in front of P3: Z1—16.6 mm, Z4—17 mm, and Z43—16.6 mm, which exceeds the height of the lower jaw in front of P3 in L. timidus (including L. t. tschuktschorum subspecies). The studied Don-hare specimens are also identified by a large diastema length: Z1 – 24.6 mm, Z4 – 22.1 mm, Z43 – 20.7 mm as well as remark-able length of the third premolar (P3): Z1 – 4.6 mm, Z4 – 3.9 mm, and Z43 – 4.1 mm [13,16].” At the same time, Don-hare and mountain hare are quite similar to each other, based on both – morphological and molecular traits.

L77: “high dental part” needs to be more specific. Reading from Gureev (1964): […] jaw with high dentition, especially the P3 (16.0-19.8 mm) and M3 (18.5-20.2 mm) [vs 13.4-14.2 mm and 15.3-18.2 in timidus). However, the teeth structure is apparently very similar to timidus?

We totally agree that species identification is an important part of the manuscript. We extended the description of the paleontological material used in this study based on your recommendations (L116-L146).

Materials and methods: 

L107-125: very impressive finds. It would be great to have some images of these specimens to accompany the text. Morphological comparison of the dentition would be interesting for the specimens with skulls, at least for P3 and M3. These should be compared with contemporary specimens of the Chukchi hare.

We extended the description of the paleontological material based on your recommendations (L116-L146). Additional Figure 2, which represents specimens considered in this study, was included in the manuscript. In the next stage of research, we are going to conduct a more detailed study of the dental morphology of the mummies of Don hares and the peculiarities of their nutrition. We plan to further study the dentition of hares using 3D scanning and will try to compare them with the modern and historical specimens of the Chukchi hare.

L173-4: If the mapping of the sequencing data was done against the whole mountain hare genome, why was only the mtDNA used for the analyses?

Thank you for the question. First, we carried out mapping on the whole Lepus timidus genome assembly (which contains mtDNA contigs) to obtain the endogenous DNA percentage. Then only endogenous DNA reads were used for de novo assembly. The nuclear genome is not described since results obtained at this stage show that additional sequencing of L. t. tschuktschorum specimens (modern and historical) is necessary.

Results:

L236: See above. How certain it is that the young specimens represent the Don hare and not the Chukchi hare? What is the identification of the specimens based on?

Thank you for your concern about species identification. We added additional information on Don-hare identification in the Material section (L116-L146). In the next step of our study, we will carry out 3D scanning of mummies of the Don-hare specimens in comparison with modern and historical specimens of Lepus timidus tschuktschorum.

Discussion:

Very nice. However, the main issue with studies on extinct species is how to obtain certainty with the species identification. I believe this is evident from the fact that different “tanaiticus” specimens cluster all around the Lepus phylogenetic tree (Figure 3). The only way to solve this would be to sequence the original type specimen, which is probably not plausible and therefore I do not require it for this study. Unfortunately, Gureev (1964) does not give much information of their age or context but considering that the type locality is in Southern Urals (Novgorod-Seversky, Voronezh region), I would not be surprised if these turn out to be something different. I think that it would be worthwhile to discuss this uncertainty in broader detail. While hybridization (L275) is plausible and known to occur, I believe that the genetic evidence shows ancient species diversity, which has not been fully understood in morphological studies and probably contains unknown species, which hopefully the nuclear genome data will someday elucidate. Unless the authors have other evidence, I would interpret that the robust animals, which cluster in the same clade with timidus (HM12682, 84) can well represent the the Chukchi hare, specimens Z4, Z5, Z43 (I hope that the letter Z has no political implications here) a species related to yarkandensis and Z1, Z3, HM126481, 83 and 85 a third species. It would be very informative to include the ages of the other “tanaicus” specimens as well as their geographic locations into the figure 3. Regardless if the authors agree with my interpretation or not, the implications of the phylogenetic tree needs to be discussed in more detail.

Thank you for the interpretation we partially included it in the Discussion section (L326-L329). However, we are not sure that the diversity of the short D-loop fragment is suitable to relate L. yarkandensis and L. tanaiticus. Unfortunately, Pleistocene-Holocene Lagomorpha species are not studied well comparing iconic megafaunal species (e.g., woolly mammoth) and deep comprehensive and collaborative international studies across Eurasia are still needed (using classic zoological and modern genomics methods).

“Z” is an abbreviation., the first symbol of a Slavic word - hare (e.g., Zając in Polish, Zajíc in Chech, Zayats in Russian).

The parsimony phylogenetic tree of the Lepus species based on nucleotide variability of 338 bp D-loop fragment (now Figure 4) was improved based on your recommendations.

Minor issues:

Abstract: remove "kilo years ago".

This part was depleted from the Abstract.

Reference:

Ognev SI (1940) Mammals of the USSR and adjacent countries. Vol 4. Akademiya Nauk SSSR, Moscow. pp. 615.

The book by Sergey Ognev was used in material description and cite

Reviewer 2 Report

In the present study, the authors did a mitogenomic analysis of the partial  (D-loop) and whole mitochondrial genome of Pleistocene Don-hare fossils from Northeastern Eurasia (Yakutia, Russia) and other ancient and  modern hare specimens and demonstrated at least two mitochondrial clades of hares in Northeast Asia. The manuscript is very straightforward and provides vital information. The manuscript is well structured and well written.

I request the authors to include some salient features observed in the mitogenomes of the Don-hares if possible.

Author Response

We would like to thank you for your consideration, valuable comments, and suggestions to improve this manuscript. Corrections in manuscript are marked by yellow. Answer is in blue.

In the present study, the authors did a mitogenomic analysis of the partial (D-loop) and whole mitochondrial genome of Pleistocene Don-hare fossils from Northeastern Eurasia (Yakutia, Russia) and other ancient and  modern hare specimens and demonstrated at least two mitochondrial clades of hares in Northeast Asia. The manuscript is very straightforward and provides vital information. The manuscript is well structured and well written.

I request the authors to include some salient features observed in the mitogenomes of the Don-hares if possible.

Thank you for your concern. Additional information with a comparison of start and stop codon presence in Don-hare and mountain hare was included in the Results section (L257-L264): “All five mitochondrial genomes of Pleistocene Don-hares have the same start and stop codons. Ten (ATP6, ATP8, COB, COX1, COX2, COX3, NAD1, NAD4, NAD4L, NAD6) of the 13 protein-coding genes used ATG as a start codon; another two (NAD2, NAD3) used ATT; and NAD5 has an ATA start codon. Nine genes (ATP6, ATP8, COX1, COX3, NAD2, NAD4, NAD4L, NAD5, NAD6) ended with a TAA stop codon, and another four (COX2, NAD1, NAD3, NAD6) use a TAG stop codon. The same start and stop codons are presented in the mitochondrial genome of the mountain hare [39].”